



# Assessing short-term climate change impacts on water supply at the Wupper catchment area, Germany

Maria Paula Lorza-Villegas[1], Rike Becker[2], Marc Scheibel[1], Tim aus der Beek[3], Jackson Roehrig[4]

[1]Department for Water Resources & Flood Risk Management, Wupper Association (Wupperverband), Wuppertal, Germany

[2]Department for Hydrometeorology & Flood Risk Management, EGLV (Emschergenossenschaft und Lippeverband), Essen, Germany

[3]IWW Water Centre, University of Duisburg-Essen, Duisburg, Germany

[4]Institute for Technology and Resources Management in the Tropics and Subtropics (ITT), Cologne University of Applied Sciences, Cologne, Germany

*Correspondence to*: Maria Paula Lorza-Villegas (pla@wupperverband.de)

**Abstract.** The occurrence of dry periods on the Wupper catchment has increased in the last decades in conjunction with the shifting of the precipitation regime. In the frame of the Horizon 2020 project BINGO (Bringing INnovation to onGOing water management), the effects of climate change scenarios on the water cycle in the Wupper catchment area were investigated. To quantify these effects, a set of hydrological models (NASIM and SWAT) has been set-up, calibrated, and validated for the

upper part of the Dhünn River catchment area - Wupper River's main tributary. This sub-catchment corresponds to one of the inflows to the Große Dhünn Reservoir (GDR), the second largest drinking water reservoir in Germany. Both models were driven with climate data from decadal predictions, which have been selected instead of IPCC-RCP scenarios, as they provide a more realistic assumption of climate variability for the next 10 years. Ten decadal members based on the MiKlip (Mittelfristige Klimaprognose - medium-term climate prediction) framework have been prepared for the time span of 2015 to

2024. Additionally, a simulation with TALSIM-NG (a reservoir-oriented hydrological model) was carried out to obtain future reservoir storage. Special focus was given to identify historical trends and compare them to future trends. Standardized Precipitation Index (SPI), Standardized Precipitation-Evapotranspiration Index (SPEI), and Standardized Runoff Index (SRI) were estimated for different seasons based on observed data to determine if they were abnormally dry or wet. SPI, SPEI, and SRI were also calculated with decadal predictions to evaluate future extreme dry periods. Uncertainties in climate data predictions are one of the greatest challenges. Observed and forecast time series were compared by means of statistical tests

in order to assess uncertainties in climate data predictions. Also, the application of two hydrological models aims to determine potential uncertainties, so that predictions are more reliable. Results indicate that SRI might be more appropriate to estimate drought periods for the study area in the frame of reservoir management - where inflow rates are of crucial importance - as this index quantifies losses in runoff formation processes. In terms of inflow rates to GDR, future changes indicate a reduction in runoff for the spring season, while an increment during winter. On the other hand, a clear change in pattern for fall and





summer seasons remains uncertain. Simulations of GDR reservoir volume with different climate scenarios show that water stress by the end of 2024 is not unlikely, so sustainable adaptation measures should be further considered. Effectively managing the GDR will become consequently more complex.

## 1 Introduction

Climate change is substantially altering not only global but also local hydrological dynamics and poses new challenges for water managers, decision, and policy-makers. In order to react adequately to these challenges, detailed knowledge about potential future hydrological changes is needed. Many climate scenarios exist, however, there is still limited practical use of such scenarios for local water managers, as predictions are often too coarse in space and time to give sufficient information on local hydrological dynamics. Main goal of the BINGO project is therefore to give practical knowledge and tools in sufficient
spatial and temporal resolution, so that local water authorities can investigate climate change impacts for their specific and sometime small-scale regions and develop adequate adaption strategies.

The local water board (the Wupper Association) is facing such challenges and is now, for the first time, able to use the decadal climate predictions of the BINGO project for a regional-specific analysis of future hydrological changes. The following study shows an example of such a small-scale climate change analysis for a local water association, and how these results can be
used to give important information for an improved regional water management.

The inflow runoff plays a central role to assess water availability in a reservoir. In this study, different climate change scenarios were evaluated in order to identify potential future dry periods on the Große Dhünn Reservoir (GDR), the second largest drinking water reservoir in Germany. In operation since 1988, GDR is located within the Dhünn River catchment area, the main tributary of the Wupper River. The Wupper catchment lies in the state of North-Rhine Westphalia, Germany, with an
area of 813 km² and a population of approximately 950,000 inhabitants (see Figure 1). The Wupper Association - responsible for water quantity management and quality of all water bodies within the Wupper catchment - operates fourteen reservoirs with a total volume of 114 Mm³, fed by 21 rivers and creeks. Alone the Große Dhünn Reservoir (GDR), with a maximum storage volume of ca. 81 Mm³, supplies drinking water for ca. 500,000 people, serving also as emergency water supplier for the city of Düsseldorf, with a contingency volume of 10 Mm³ (BINGO D3.2, 2016). Due to GDR's regional significance, it is
of crucial importance to assess future water security.

The occurrence of dry periods in the Wupper catchment has increased in the last decades together with the shifting of the rainy season (BINGO D3.1, 2016). This has important impacts on hydrological processes controlling the inflow to the reservoir and future water availability in the Wupper region. To be able to explain predicted changes in inflow rates, this study first analyzes past climate to detect historical trends. It then uses the hydrological models SWAT (Arnold et al., 2012) and NASIM (by
Hydrotec) to predict future variations on inflow runoff.





River regimes can be impacted by changes in precipitation, combined with greater atmospheric evaporative demand (AED) (Vicente-Serrano et al., 2017), thus, affecting reservoir levels. In the frame of this research, decadal predictions were also used to detect future potential changes in river flow controlling variables, such as precipitation, temperature, and evapotranspiration. The final aim of this study is to determine upcoming occurrences of hydro-meteorological drought and climate change impacts on inflow rates in the near future.

## 2 Study area

Mean annual precipitation (MAP) in the Wupper catchment varies from about 700 to 1400 mm. On account of the higher rainfall amounts in the upper parts of the basin, the construction of large reservoirs started towards the end of the 19th century. MAP has remained relatively constant with respect to the weather normal distribution. However, mean monthly precipitation during the 20th century has shown a shifting of the rainy season from spring (April) to summer (June-July). As a consequence, reservoirs are being filled up in summer rather than in spring, lacking the adequate temperature for ecological flow and affecting water quality (BINGO D3.1, 2016). Summer months in the Wupper catchment area have been relatively dry. Furthermore, most of the summer precipitation events does not reach the reservoir on account of evapotranspiration and interception losses. This alone does not lead to water scarcity stress. Water availability issues arise at the reservoir when a dry summer follows a dry winter-spring period (or several consecutive dry winter-spring periods). In general, filling up of the reservoirs has not relied on summer precipitation.

The study area for this research corresponds to the upstream region of the inflow to GDR (upper Große Dhünn sub-basin, see Figure 1), i.e., the drainage area upstream of Neumühle hydro-meteorological station. Since GDR is a drinking water reservoir, the areas in its surroundings are protected. This sub-catchment has therefore remained rather natural, being mostly covered by deciduous and coniferous forest. This basin has an area of around 23 km².

Since 1988, three dry spells have been registered at GDR, where the reservoir water level achieved a critical threshold for several, consecutive days. Based on adequate limnological conditions, the critical reservoir storage level is defined as 35 Mm³. Table 1 presents the duration of each dry spell as well as the corresponding storyline between relevant stakeholders and implemented response measures.

Projected changes in precipitation are highly variable between regional climate models (DWD, 2018). According to DWD (2018), an increasing or decreasing in precipitation up to 26 % can be expected for Germany. Climate projections for Germany indicate also an increase in surface air temperatures by 2.8-5.2° C for the RCP8.5 scenario by the end of the century (2071-2100, compared to 1971-2000), likely affecting precipitation regimes.





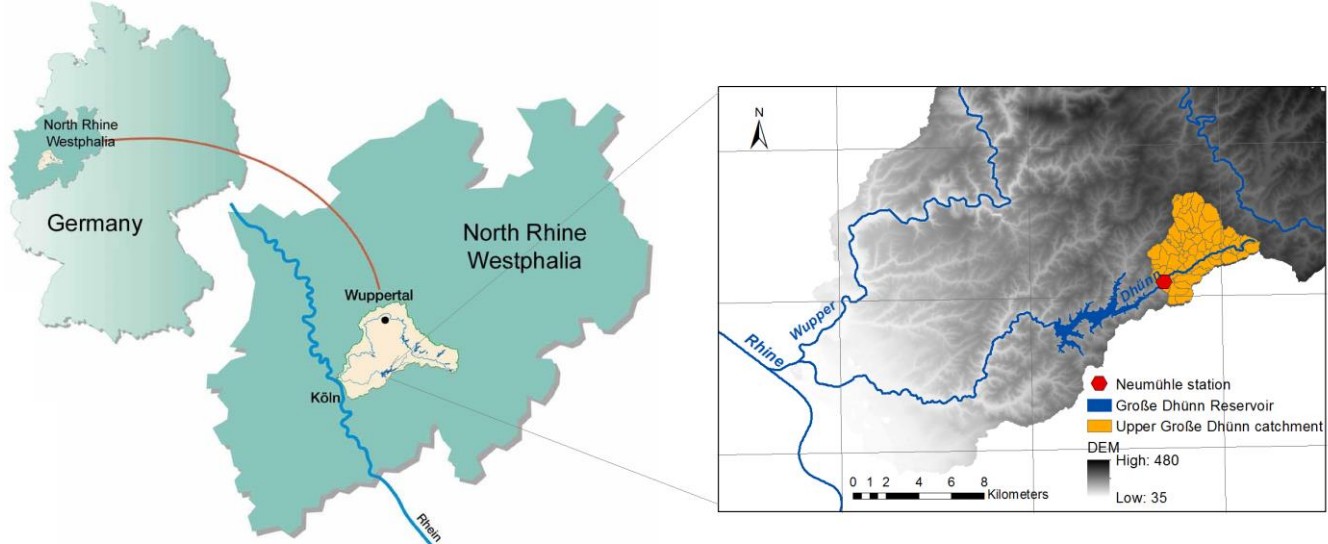

Figure 1: Location of the Wupper River basin and the study area: the upper Große Dhünn sub-catchment

## 3 Data and methods

Meteorological drought refers to the deviation from average conditions of meteorological parameters which causes drying of the surface. This type of drought is typically region-specific since the atmospheric conditions in different areas are highly variable in time and space (Liu et al., 2013). Hydrological drought takes place when low water supply turns evident, particularly in streams, reservoirs, and groundwater levels, usually after many months of meteorological drought (NOAA, 2020). The severity and frequency of hydrological droughts are often defined on a basin-scale. Hydrological droughts are normally out of phase with the occurrence of meteorological droughts since it takes longer for precipitation shortfalls to be evident on the other components of the hydrological system, such as reservoir levels (NDMC, 2020). While the effects on the agricultural sector are almost immediately noticeable, the impacts on water supply are mostly only discernible after many months, which makes the quantification of impacts more complex (NDMC, 2020).

In some regions, hydrological droughts present longer duration and higher severity in comparison to meteorological droughts (Vicente-Serrano et al., 2017). This can be accounted for the greater AED resulting from a warmer climate (Vicente-Serrano et al., 2014), increased urban water demands, and the expansion of irrigated and urban areas (Lorenzo-Lacruz et al., 2013).



**Table 1: Dry spell durations at the Große Dhünn Reservoir and implemented response measures**

| Event | From | To | Duration (days) | Total duration (days) | Storyline |
|---|---|---|---|---|---|
| 1 | 03.09.1991 | 21.12.1991 | 109 | 109 | - Not known effects; besides, the time where the reservoir volume was below the threshold was short. |
| 2 | 20.04.1996 | 14.02.1997 | 300 | 561 | - In May 1995, the volume started to sink. Afterwards, there was no recovery phase, i.e., the winter 95-96 was dry.<br>- This was the first time that there was no recovery phase. |
| | 14.08.1997 | 08.04.1998 | 237 | | - Then, the maximum reservoir level was only a "small" peak. This was the first time where the maximum level after a winter was so low (March 1997). |
| | 22.08.1998 | 15.09.1998 | 24 | | - The discussion related to dry periods and corresponding measures started to take place between relevant stakeholders. However, after this "double" dry years (i.e., 1996 - 1998), several normal years followed and the discussion was set aside (until the next extreme dry event). |
| 3 | 22.05.2014 | 09.01.2015 | 232 | 395 | - This has been the most extreme event registered so far.<br>- It was the first time where the water suppliers* had to give up 20 % of their contractual raw water withdrawal as response measure. |
| | 01.07.2015 | 11.12.2015 | 163 | | - Negative impacts were avoided by response measures, such as temporary reduction of low flow augmentation for ecological flow in downstream areas. |

* Water treatment from GDR as well as its later distribution to the supply network is responsibility of the following water suppliers: Water Supply Association Rhine-Wupper (Wasserversorgungsverband-Rhein-Wupper - WVV, contractual raw water withdrawal = 5.7 Mm³) and Bergische Drinking Water Association (Bergische Trinkwasserverband - BTV, contractual raw water withdrawal = 36.3 Mm³)

**3.1 Data**

**3.1.1 Observed data**

Ground observations at Neumühle station are available since 1990. Time series of precipitation, temperature, reference evapotranspiration (ETo), and discharge were used to analyze historical climate in the study area as well as to evaluate decadal predictions skills. ETo was estimated with the Penman-Monteith method based on observed daily mean air humidity and

temperature, wind velocity, and sunshine hours. Temperature time series of Buchenhofen station located in the city of Wuppertal were used in order to identify past trends in the last decades. This station was selected since it provides long-term continuous temperature time series (since 1948). In addition, GDR storage time series are available since 1988 and used for comparison with simulated volume with different scenarios.



### 3.1.2 Decadal predictions

Decadal climate predictions are a relatively new field which aims to simulate the climate response to future anthropogenic forcing as well as future evolution (from the present) of the climate due to internal climate variability (Marotzke et al., 2016 cited in Rust et al., 2018). This approach differs from the one implemented in climate projections, whose focus is on the climate response to anthropogenic forcing, and the impacts of internal climate variability are meant to be nullified via multi-decadal climate model integrations (Rust et al., 2018).

Therefore, decadal predictions have been selected for this research instead of IPCC-RCP scenarios, as they provide a more realistic assumption of climate variability for the next 10 years. Ten decadal members (also refer to as "realizations") based on the MiKlip (Mittelfristige Klimaprognose - medium-term climate prediction) framework have been prepared for the time span of 01 January 2015 to 31 December 2024.

According to Rust et al., 2018, the most accurate prediction will be the ensemble mean, which has been employed within the
scope of this study. Since the decadal members providing the largest span (i.e., the realizations furthest above/below the ensemble mean) are also of interest, they were determined based on resulting simulated discharge (total accumulated volume) at Neumühle station (BINGO D3.4, 2018).

In the frame of BINGO, the decadal predictions have been retrieved from DECO. DECO is a plug-in for data extraction and conversion developed within and for BINGO by Rust et al., 2018 (freva.met.fu-berlin.de). All variables are available at 12-km
spatial resolution, at daily and 3-hourly time steps. Specifically, the retrieved meteorological variables for this study are precipitation, air temperature and humidity, short-wave solar radiation, and wind velocity. The decadal prediction ensemble (i.e., the set of all separate realizations) has been bias-corrected using the Cumulative Distribution Function transformation (CDFt) method. Further details on the implemented bias-correction method as well as the procedure to generate the decadal predictions data can be found in Rust et al., 2018.

### 140   3.1.3 Runoff and reservoir volume simulations

A set of hydrological models in NASIM and SWAT has been set-up, calibrated, and validated for past conditions using ground data for the upper part of the Dhünn catchment area. Model performance was assessed with different statistical metrics, such as relative volume error, coefficient of determination, and Nash-Sutcliffe Efficiency (BINGO D3.3, 2016; BINGO D3.4, 2018).

NASIM and SWAT are physically, lumped, hydrological models based on the water balance equation. While NASIM offers
high level of detail for modelling of complex urban areas and hydraulic structures as well as the possibility of entering precipitation time series at fine temporal resolution (e.g., minutely data), SWAT enables to study long-term impacts offering





a huge variety of input and output variables including different soil properties, vegetation, and land management practices. For most calculations, SWAT operates on a daily time step.

For NASIM, input time series are precipitation (in any temporal resolution), daily mean air temperature, and daily ETo. ETo
must be previously estimated and then entered into NASIM model. As mentioned, ETo was calculated with the Penman-Monteith method in the frame of this research. For this, parameters such as short-wave solar radiation, wind velocity, air humidity, and air temperature are required. In turn, SWAT performs an internal calculation for the estimation of ETo based also on the Penman-Monteith method. Simulated inflow to GDR at Neumühle station with NASIM and SWAT was obtained from 01 January 2015 and 31 December 2024 and compared to observed discharge.

In the frame of "BINGO D3: Special report: impacts of long-term climate change effects on typical water problems in Europe" (BINGO D3, 2019), a simulation with TALSIM-NG was carried out in order to obtain reservoir storage from 01 January 2015 to 31 December 2024 (BINGO D3.4, 2018; BINGO D3, 2019). This simulation was performed with RCP4.5, RCP8.5, and the decadal members providing the largest span (i.e., realizations 1, 7, and 9). As mentioned, realizations providing the largest span (max, min, and mean) were determined based on simulated discharge at Neumühle and are considered representative for
future climate scenarios (BINGO D3.4, 2018). TALSIM-NG (by SYDRO Consult) is a physically-based, reservoir-oriented water balance model, which allows the simulation of complex operational rules taking into account raw water extraction and service water utilized for flood management and ecological flow regulation.

### 3.2 Statistical tests

Decadal predictions are available for the time span of 01 January 2015 to 31 December 2024, providing almost a complete
six-year overlapping period with observed time series to the present day. In this manner, it is possible to evaluate the skill of forecast series as a way of quantifying uncertainty. Unpaired two-sample t-test, Welch t-test, and Mann-Whitney-U-test were applied in order to assess the similarity between the central values (i.e., mean or median values) of observed and forecast data.

In the frame of this research, the significance level (alpha) of 0.05 was set-up as a typical threshold for statistically significant changes with a corresponding confidence interval of 0.95 (95 %). For all tests, if the p-value is less than or equal to the
significance level (p-value ≤ 0.05), the null hypothesis, $H_0$, is rejected.

The unpaired two-sample t-test is a parametric test and is used to compare the mean values of two independent samples, i.e., two unrelated or unpaired data sets. The unpaired two-sample t-test is also commonly applied to continuous variables that are normally distributed (Xu et al., 2017). Thus, for the application of this test, the following assumptions must be met:

✓ Both data sets are normally distributed
✓ The variances of both data sets are equal



The two-sample F-test is used to check the equality of the variances of the two samples. If this condition is not fulfilled, the Welch t-test can be used instead (Stagge et al., 2015). In turn, the Shapiro-Wilk test determines whether a sample follows a normal distribution. If one of the data sets is not normally distributed, the non-parametric Mann-Whitney-U-test can be applied instead of the t-test to assess similarity between the median values of two data sets (Stagge et al., 2015). The null hypothesis,

$H_0$, for this test establishes that both samples have the same median.

To identify past trends, the non-parametric Mann-Kendall test was applied, where the alternative hypothesis, $H_A$, states that there is a monotonic trend. Hence, statistically significant monotonic trends are defined as those samples having p-values ≤ 0.05.

### 3.3 Drought and climate indices

In the scope of this study, three drought indices were calculated, namely, SPI (Standardized Precipitation Index), SPEI (Standardized Precipitation-Evapotranspiration Index), and SRI (Standardized Runoff Index).

SPI (McKee et al., 1993) has been widely used in many studies on account of its simplicity since only takes monthly precipitation as input. SPI serves to identify precipitation surpluses and deficits (DWD, 2017). Precipitation data are fitted typically to a gamma distribution and then transformed to a normal distribution. SPI values are interpreted as the number of

standard deviations by which the anomaly deviates from the long-term mean (Keyantash, 2016). Negative values indicate drier periods than average conditions, whereas positive values indicate wetter periods than normal; values close to zero reflect average conditions (see Table 2).

The Standardized Precipitation-Evapotranspiration (SPEI) is also a climatological drought index based on a monthly climatic water balance (Vicente-Serrano et al., 2010). The monthly climatic water balance is defined here as the difference between

precipitation and ETo, and thus, incorporates changes in AED (Vicente-Serrano et al., 2017). Similar to SPI, SPEI uses a normalization approach, where a log-logistic probability distribution is typically fitted to the empirical distribution of accumulated water balance over a selected aggregation period (Meresa et al., 2016). SPEI is better suited for climate change analysis than SPI since it takes into account not only precipitation but also temperature. Like SPI, negative values indicate drier periods than normal.

McKee et al., 1993 suggest that the procedure to determine SPI can be used with other variables related to drought, such as streamflow or reservoir storage. The Standardized Runoff Index (SRI, introduced by Shukla, 2008) is calculated similarly to SPI using discharge instead of precipitation. SRI uses monthly streamflow data as its main input. Since streamflow depends in turn on other components of the hydrological cycle (infiltration, soil moisture, ground water, etc.) as well as basin characteristics (e.g., size, topography, land use, and soil type), SRI incorporates hydrological processes which determine

seasonal lags in the influence of climate on runoff (Shukla, 2008).





In this research, hydrological drought was quantified with SRI, whereas meteorological drought was determined with SPI and SPEI. The three indices were calculated with the Python package *caeli* (Roehrig and Lorza-Villegas, 2020).

**Table 2: Drought classification for SPI, SPEI, and SRI (Lloyd-Hughes and Saunders, 2002)**

| Index value | Drought category |
|---|---|
| > 2.00 | Extremely wet |
| 1.50 to 1.99 | Severely wet |
| 1.00 to 1.49 | Moderately wet |
| 0.00 to 0.99 | Mildly wet |
| 0.00 to -0.99 | Mildly dry |
| -1.00 to -1.49 | Moderately dry |
| -1.50 to -1.99 | Severely dry |
| < -2.00 | Extremely dry |

Climate indices are diagnostic tools used to characterize the state of the climate system (CDIAC, 2012). In order for individuals, countries, and regions to calculate the indices in exactly the same way such that their analyses will fit seamlessly into the global picture, the joint Expert Team (ET) on Climate Change Detection and Indices (ETCCDI) has coordinated an international effort to develop, calculate, and analyze a suite of indices (Karl et al., 1999; Peterson et al., 2001). With this purpose, 27 core climate change indices have been established (ETCCDI, 2020). Among them, the number of summer days can be used to identify a raising trend in temperatures. Summer days are defined by the number of days with maximum air temperatures above 25° C (ETCCDI, 2020). In the context of this study, warmer summers bring as a consequence greater water demand and higher evapotranspiration rates, thereby affecting reservoir levels.

## 4 Results and discussion

### 4.1 Historical climate analysis

The World Meteorological Organization (WMO) defines *climate normals* as 30-year reference periods of climate observations. Recent or current observations can be contrasted against climate normals, serving as a benchmark for comparison (WMO, 2017). The time interval from 1961 to 1990 is often used as the historical control period (GERICS, 2020). For this analysis, Buchenhofen station (located in the city of Wuppertal) was selected as it provides continuous temperature time series since 1948. The ongoing climate normal (1991-2020, not yet completed) was compared to the control period (1991-1990).

Figure 2a illustrates the anomalies depicted as values above or below the long-term mean temperature of the control period. Linear trends of the control period and the ongoing climate normal are also displayed in Figure 2a. For both periods, there is





an increasing trend in temperature, showing a significant climate signal. The non-parametric Mann-Kendall test was able to prove the significance of both positive trends (p-value ≤ 0.05). Table 3 summarizes the linear trends of the individual time intervals, where 1961-1990 shows the greatest warming. For the entire period of 1948-2020, there has been a temperature rise of approx. 2° C in total, with an average increase of 0.03° C per year.


**Table 3: Summary of the linear trends for each individual time period**

| Period | Trend [° C] | |
| --- | --- | --- |
| | Total | Per year |
| 1961-1991 | 1.27 | 0.04 |
| 1991-2020 | 0.73 | 0.02 |
| 1961-2020 | 2.00 | 0.03 |

The two-sample t-test was applied for both time periods (1961-1990 vs. 1991-2020) in order to test their similarity. Results show a statistically significant difference between both data sets (p-value = 2.655e-07). The control period presents a mean air temperature of 9.4° C, whilst for the ongoing climate normal, the mean temperature is 10.5° C. This implies that the ongoing climate normal has been warmer than the control period (Figure 2b), depicting also more positive anomalies (see Figure 2a).

A comparison with previous years also shows that the annual number of summer days are steadily increasing (see Figure 3). Particularly, the highest measured temperature was recorded in the summer of 2018, during which an extreme flood event caused by heavy rain took place in the city of Wuppertal.

As the same pattern can be observed in other stations located elsewhere in the Wupper catchment area, historical climate analysis indicates an overall increment in mean temperatures. The temperature rise has led to an increase in convective heavy

precipitation events, particularly in summer, further impacting the monthly and seasonal precipitation regime. This is consistent with Hartmann et al., 2013, who state that an increase in precipitation extremes are in line with a warmer climate. In conjunction with the subsequent greater water demand and rise in AED, river streamflow (i.e., inflow runoff to GDR) will continue to be affected in the future.

With the aim of assessing impacts of seasonal changes on flow processes (precipitation, evapotranspiration, and inflow runoff

rates), SPI, SPEI, and SRI were calculated with observed data from Neumühle hydro-meteorological station. Each index was determined for the four seasons (i.e., winter, spring, summer, and fall), using a three-month aggregation (see Figure 4). The analysis at seasonal scale is considered to be adequate for reservoir management of GDR.





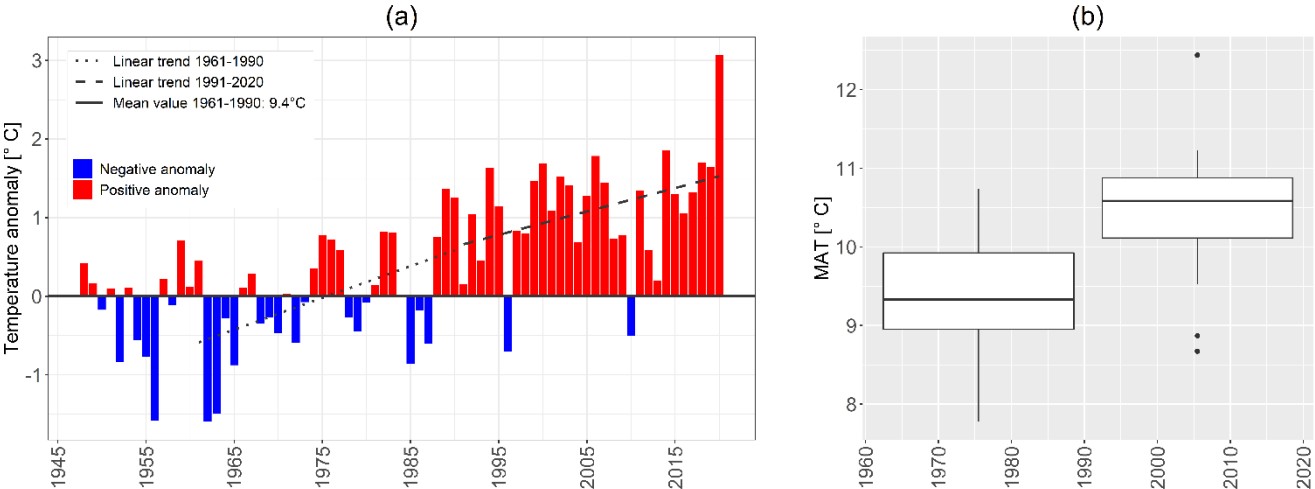

**Figure 2: Long-term temperature changes at Buchenhofen station: a) temperature anomalies above or below long-term mean temperature of the control period; b) Mean Annual Temperature (MAT) divided into two periods (1961-1990 and 1991-2020)**

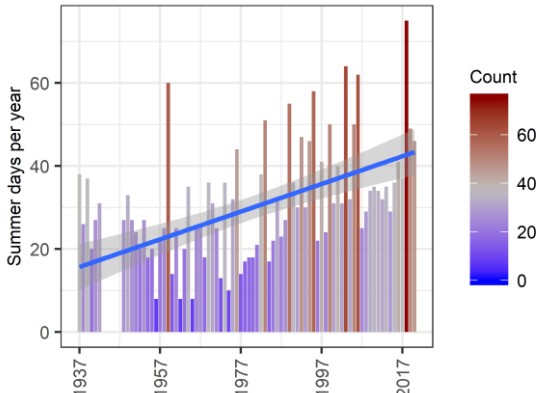

**Figure 3: Number of summer days per year at Buchenhofen station**

*Winter*: all indices show the same increasing trend, indicating that the winter season has become more humid in the last years. SPI and SPEI show the strongest increasing trend in comparison to SRI. Since in winter there is less interception as well as lower evapotranspiration rates, SPI and SPEI trends are a response of almost only precipitation. The catchment accumulates the fallen precipitation, which is not entirely transformed into surface runoff by the end of the winter season. Thus, the increasing SRI trend is much more attenuated (less steep). This slower hydrological response suggests that even an abundant precipitation regime in winter will not necessarily be reflected in river discharge.

*Spring*: SPI does not present a clear trend. SPEI in turn shows that springs have become drier due to higher evapotranspiration rates caused by increasing temperatures. SRI shows an even stronger negative trend than SPEI, indicating that in spring, SRI is more responsive to precipitation shortages. This suggests that the decrease of precipitation in spring overtime has a greater





impact on runoff generation on account of other components of the hydrological cycle (e.g., infiltration and groundwater recharge).

*Summer*: SPI and SPEI do not show a clear trend; SRI, however, presents a decreasing trend. Discharge is proportional to the
state of catchment wetness. The negative SRI trend by the end of spring implies a low catchment wetness at the beginning of summer. This, in turn, leads later on to a decreasing SRI trend in the summer season. Even though there is no clear SPI or SPEI trend, a sequence of moderate to severe dry summers have occurred in the last three years. In terms of the indices values, SPEI shows more extreme values than SPI on account of higher evapotranspiration rates caused by increasing temperatures in summer. It can be then inferred that the decrease in SRI trend in spring is mainly on account of evapotranspiration rates,
whereas in summer, the negative SRI trend takes place due to catchment wetness by the end of the spring season.

*Fall*: all three indices show a negative trend, whereas SRI presents the strongest slope. Following the same pattern as in summer, this can be the result from initial drier conditions at the beginning of fall due to decreasing SRI values at the end of the precedent season.

By negative trends, SRI always depicts the strongest slopes, whereas by positive trends (i.e., more than average water
availability), SRI trend's slope is not as steep. SRI can therefore be considered a more conservative index than SPI or SPEI. As SRI quantifies losses in runoff formation processes, this index might be more appropriate to estimate drought periods for the study area in the frame of reservoir management, where inflow rates are of crucial importance. It can be inferred that this is also the case for other reservoirs in the Wupper catchment area.

**4.2 Future changes in precipitation, temperature, ETo, and discharge**

To assess potential future changes in precipitation, temperature, ETo, and discharge values in the upper Große Dhünn catchment, decadal predictions were analyzed for the period 2015-2024 and compared to the historical reference period. The historical reference period is defined here as the time interval from 01 January 1990 to 31 December 2014. Figure 5 illustrates observations and the forecast of all 10 decadal prediction ensemble members at seasonal scale. Variations between the 10 model realizations lie within the inter-annual variability of past observations. Clear and statistically significant trends in future
changes of precipitation, temperature, and ETo are therefore not detectable.






**Figure 4: Seasonal SPI, SPEI, and SRI with observed data. Time period: 1990–2020**




Based on the decadal predictions, precipitation (P) is forecasted to slightly reduce in future (see Figure 5a). Strongest reductions
are estimated for the winter months (DJF), which show a mean decrease of approx. 100 mm. While summer and fall months
present a small reducing trend (approx. 30 to 40 mm per season), spring shows no future changes.

Predictions of future temperatures (tas) show that mean summer and fall temperatures are expected to increase by approx. 0.5°
C and 1.5° C, respectively, while winter and spring temperatures do not reveal significant changes (see Figure 5b). The summer
trend is less pronounced than the fall trend.

Similarly to the prediction of temperature changes, decadal members forecast an increase in future ETo during summer months
(see Figure 5c). However, the variability between the realizations is high and no clear summer trend can be estimated. Overall
changes in ETo rates are low for spring and fall. Winter months show a slight reduction in ETo, however, ETo does not play
a significant role in winter in any case. Therefore, slight changes in ETo in winter are not relevant.

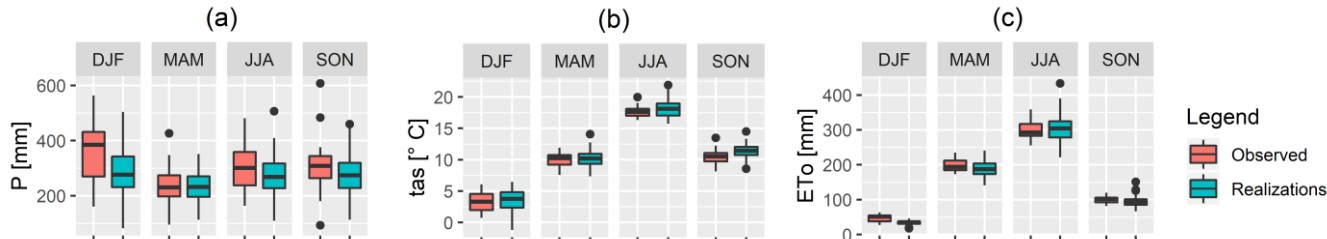

**Figure 5: Observed data (1990-2014) and decadal predictions of all ensemble members (2015-2024)**

To estimate potential future changes in discharge rates and hence changes in inflow to GDR, the two hydrological models
NASIM and SWAT were used. The time series provided by the decadal predictions were taken as input data to simulate future
changes in discharge dynamics. In line with the previous analysis, observed discharge from 1990 to 2014 was compared with
simulated discharge (Q) by NASIM and SWAT for the period 2015 to 2024 (see Figure 6). Both models show similar trends
in their runoff estimations and predict decreasing discharge rates in winter (approx. 0.1 to 0.2 m³ s⁻¹) and increasing discharge
in summer and fall (approx. 0.1 m³ s⁻¹ and 0.05 m³ s⁻¹, respectively). Only for spring months (MAM), NASIM predicts a slight
increase of mean runoff, whereas SWAT estimates a slight decrease of mean runoff rates. Even though the predictions of both
models for spring lie within the interquartile range of the historical data, a significant increase or decrease compared to past
conditions cannot be stated. Overall, spring and fall seasons show the least changes in runoff rates, and in general, SWAT
generates less runoff along all seasons. Nonetheless, there is a general good agreement between both models, showing that
model uncertainty is relatively small, and the use of hydrological model ensemble is valid for further analysis. This is an
indication that selecting an appropriate hydrological model is not as critical as the quality of the input data.

It should be noted that the variability in the ensemble predictions of both models is higher than the inter-annual variability of
observations, making a clear prediction of future discharge trends difficult. For this reason, ensemble studies are often done





with the aim to provide an ensemble mean value of climate variables. However, taking the ensemble mean of all 10 realizations reveals the important fact that this leads to a significant underestimation of the seasonal discharge variability (see Figure 6b). Maintaining the spread of the entire 10 ensemble members represents better the observed variability (see Figure 6a). The loss of potential variability in discharge data is especially important when ensemble means are taken as basis for water management decision (e.g., reservoir operational rules and water allocation decisions).

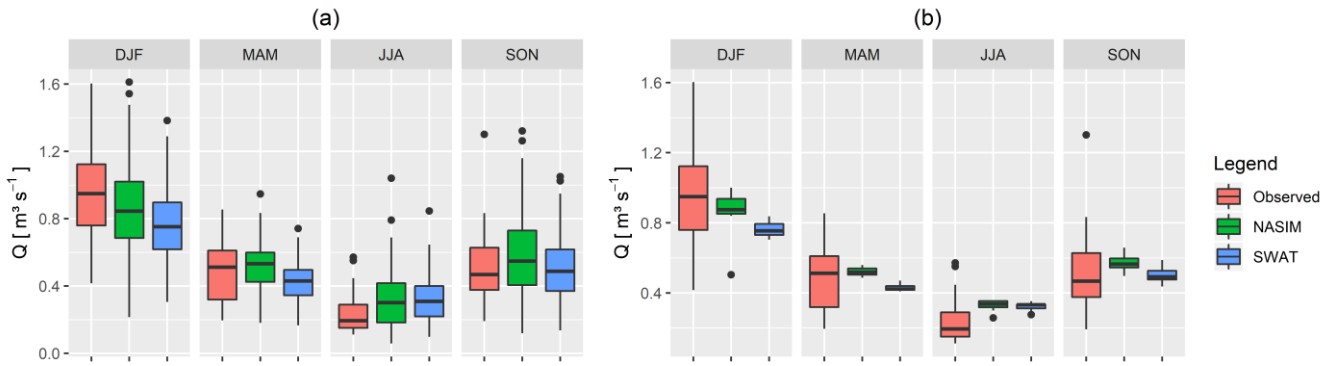

**Figure 6: Observed seasonal discharge rates (1990-2014) and future discharge rates (2015-2024): a) simulated seasonal discharge rates of all ensemble members are preserved; b) simulated seasonal discharge rates are averaged over all ensemble members (i.e., ensemble mean)**

As with decadal predictions, where ensemble mean aims to reduce model uncertainties (Rust et al., 2018), it seems reasonable
to apply the same principle to resulting discharge by two different hydrological models (i.e., hydrological model ensemble). Past studies have shown that uncertainty can be reduced even with simple ensemble techniques such as arithmetic mean (Bormann, et al., 2007). Moreover, hydrological model ensemble can yield better results than each individual model or the best one of them (Georgakakos et al., 2004). Simulated discharge of both hydrological models (NASIM and SWAT) was hence averaged for further analysis.

Today, after the creation of the decadal predictions for the period 01 January 2015 to 31 December 2024 in the frame of BINGO, it is possible to compare forecast with observed data for almost six overlapping years (i.e., 2015 to 2020) and test their similarity. Table 4 shows the seasonal mean values for both ensemble members as well as for the observations. Whenever the Mann-Whitney-U-test was applied, the median value is displayed instead. In addition, it shows the results of the applied tests, indicating if the ensemble members' central values are significantly different from observed central values (yes =
ensemble members' central values are significantly different with a p-value ≤ 0.05; no = ensemble members' central values are not significantly different at a 95 % confidence interval). The hydrological model ensemble of NASIM and SWAT was used for discharge comparison.





Test statistics confirm an overall good agreement between ensemble members and observations. Statistically significant differences are only detected for temperature (fall season) and discharge (summer season). For temperature in fall, there is a slight overestimation of the members mean value. In contrast, for summer discharge, there is a significant overestimation of the models mean value with respect to observed mean. This overestimation represents approx. 2.46 times the observed mean. This suggests poorer skills of decadal members for summer discharge prediction. The effects of these uncertainties are transferred to both hydrological models, which show similar results. Therefore, future changes detected in summer should be interpreted with caution.

**Table 4: Central values and tests results between decadal predictions and observations. Time period: 2015-2020**

| Variable | Season | Observed | Members | Performed Test | p-value | Observed vs. Members: significantly different? |
|---|---|---|---|---|---|---|
| P [mm] | DJF | 406.60 | 319.22 | Mann-Whitney-U-test | 0.4206 | No |
| | MAM | 226.92 | 242.48 | Welch-t-test | 0.4675 | No |
| | JJA | 246.94 | 286.74 | Welch-t-test | 0.3470 | No |
| | SON | 290.78 | 285.96 | Welch-t-test | 0.9154 | No |
| tas [° C] | DJF | 2.61 | 2.74 | t-test | 0.8155 | No |
| | MAM | 8.93 | 9.09 | t-test | 0.6498 | No |
| | JJA | 17.23 | 16.92 | t-test | 0.2524 | No |
| | SON | 9.27 | 10.38 | t-test | 0.0001 | Yes |
| ETo [mm] | DJF | 43.10 | 33.65 | Mann-Whitney-U-test | 0.1508 | No |
| | MAM | 205.38 | 187.12 | t-test | 0.0865 | No |
| | JJA | 321.46 | 299.14 | t-test | 0.1574 | No |
| | SON | 101.06 | 90.81 | t-test | 0.0798 | No |
| Q [m³ s$^{-1}$] | DJF | 0.96 | 0.76 | Welch-t-test | 0.2426 | No |
| | MAM | 0.40 | 0.48 | Welch-t-test | 0.0938 | No |
| | JJA | 0.13 | 0.32 | Welch-t-test | 0.0072 | Yes |
| | SON | 0.31 | 0.53 | Welch-t-test | 0.0664 | No |

Observed and simulated individual discharge rates were compared for the validation period (2015-2020, see Figure 7a). Such a comparison helps to assess if the predicted decreasing runoff rates in winter and increasing runoff rates in summer and fall could in fact be observed or if model predictions over- or underestimated future discharge rates. Figure 7b shows that except for a clear overestimation of ensemble summer discharge rates, the predicted discharge volumes lies within the range of observed variability. Significant over- or underestimation by the decadal predictions can therefore be ruled out.




Nevertheless, based on the results of the six overlapping years, it can be stated that the models tend to overestimate the discharge reduction during the winter season, while overestimating discharge rates during summer and fall seasons. The best agreement between observed and simulated discharge takes place in spring. This is an important finding for the water
management and planning in this region.

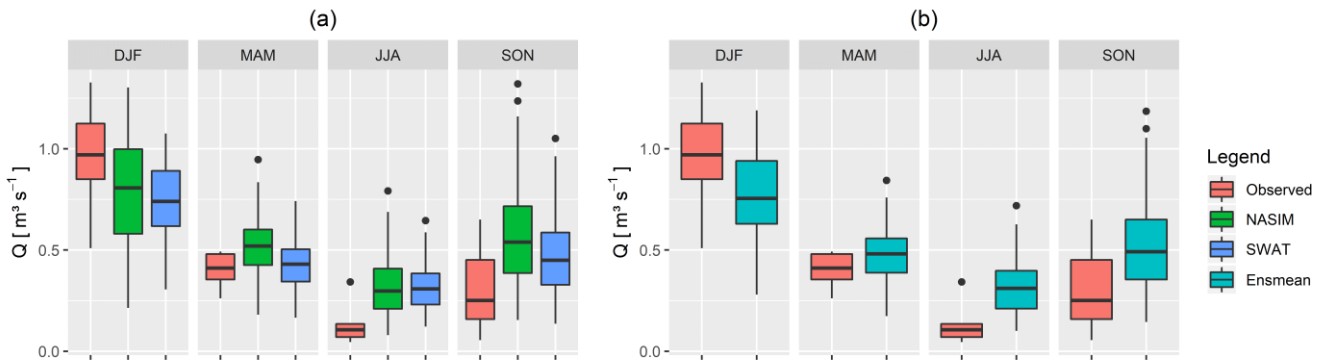

**Figure 7: Observed seasonal discharge rates and estimated discharge rates. Time period: 2015-2020: a) observed vs. simulated discharge by NASIM and SWAT; b) observed vs. simulated discharge (hydrological model ensemble)**

### 4.3 Future changes in drought indices and reservoir storage

This section addresses standardized anomalies of monthly precipitation, climatic water balance (P - ETo), and streamflow series. Following the methodology to assess past trends, SPI, SPEI, and SRI were calculated with input data from decadal predictions (time span: 2015 to 2024, see Figure 8). SRI was calculated with the resulting ensemble discharge from NASIM and SWAT.

Since there is almost a six-year overlapping period (2015 to 2020, corresponding to more than half the sample size of decadal
members), predicted seasonal indices are expected to show similar trends compared to observed seasonal indices. For winter, all indices show the same increasing trend, with SRI presenting the least steep slope, which is in line with past trends (see Figure 4), and with the fact that decadal predictions tend to underestimate discharge rates during the winter season (see Figure 7). For spring, all indices depict a decreasing trend with a strong negative slope. An increasing of drier springs is also consistent with historical trends.


Predicted SPI and SPEI for summer depict clear negative trends, differing slightly from historical data, where SPI and SPEI do not present clear trends (see Figure 4). This, however, is in line with the moderate to severe dry summers of the last three consecutive years. On the other hand, SRI does not show a clear trend for the summer season. This could be explained by uncertainties derived by the hydrological models on account of input data from the decadal members (see Sect. 4.2). In contrast





to past trends, fall presents positive trends depicted by all indices, which is not consistent with past data. This should be further investigated.

Despite the short sample size of the realizations, it can be stated that the trends of decadal predictions are consistent with observed trends in winter and spring, while summer and especially fall do not show a good agreement. It should be noted that the low agreement in summer and fall do not compromise as much the usability of decadal predictions in the frame of reservoir

management, since the critical seasons for filling up the reservoir are winter and spring.

Reservoir volume at GDR was simulated with TALSIM-NG from 2015 to 2024, considering current operational rules (BINGO D3.4, 2018; BINGO D3, 2019). The results displayed in Figure 9 show that realizations 1 and 9 present the best agreement between observed and simulated reservoir storage. This is expected since decadal predictions resemble climate patterns more accurately than climate projections (Kadow et al., 2017). All data sets present a negative trend by the end of 2024, with

realization 9 being the most critical scenario. Given the good agreement between observed and simulated reservoir storage for the validation period of 2015 to 2020, water stress by the end of 2024 is not an unlikely scenario. Possible adaptation measures were also determined in the frame of the BINGO project (see e.g., Strehl et al., 2021) and include: temporary reduction of low flow augmentation for ecological flow regulation in downstream areas, water transfer from another reservoir, use of alternative water sources, and reduction of water consumption. Of all measures, reduction of low flow augmentation has already been

implemented and has shown the potential to significantly reduce risk. However, its implementation is only possible to a limited extent in conjunction with complex water monitoring. It also generates opportunity costs and conflicts with ecological needs in downstream areas. These aspects should be considered to enable a long-term and sustainable implementation in the frame of climate change.



**Figure 8: Seasonal SPI, SPEI, and SRI with decadal predictions. Time period: 2015–2024**


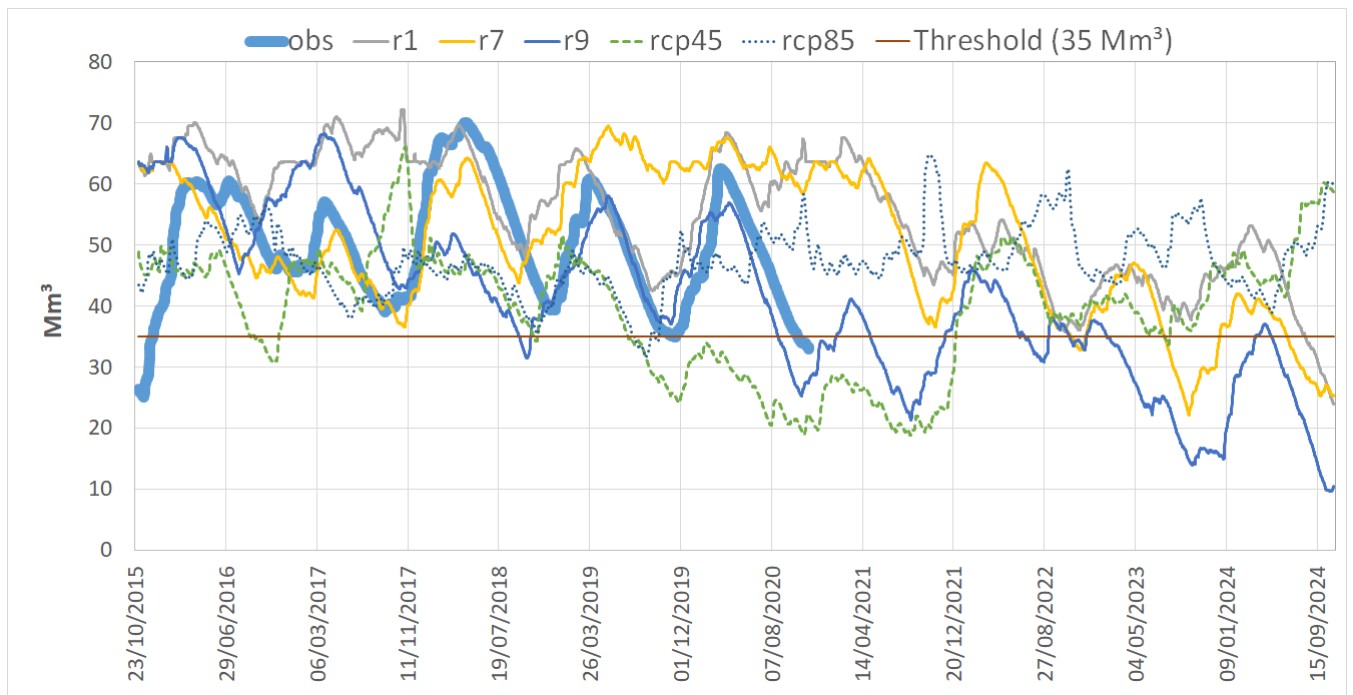

**Figure 9: Reservoir storage volume – observed vs. decadal members vs. RCP4.5 and 8.5 (2015 – 2024)**

## 5 Conclusions

Historical climate analysis indicates an overall increment in mean temperatures in the Wupper catchment area. Especially in
summer, a future increase in temperature is likely to cause more frequent and intense convective rainfall events, which may
not reach the reservoir. Higher evapotranspiration rates would cause summers to become inherently drier, which in conjunction
with the subsequent greater water demand, will have a negative impact on water availability at the reservoir, especially in case
of preceding dry springs.

To assess impacts of seasonal changes on flow processes (precipitation, evapotranspiration, and inflow runoff rates), SPI,
SPEI, and SRI were first calculated with observed data. The indices show that winters have become more humid, while springs
and fall drier. For summer, SPI and SPEI do not show a clear trend, whereas SRI presents a decreasing trend. It can be stated
that SRI might be more appropriate to estimate drought periods for the study area in the frame of reservoir management -
where inflow rates are of crucial importance - as this index quantifies losses in runoff formation processes. For past trends,
SRI shows the highest sensitivity to precipitation shortages. It can be inferred this is also the case for other reservoirs in the
Wupper catchment area.





SPI, SPEI, and SRI were also calculated with decadal predictions. Since there is almost a six-year overlapping period (2015 to 2020, corresponding to more than half the sample size of decadal members), predicted seasonal indices are expected to show similar trends compared to observed seasonal indices. Decadal predictions present the best agreement for winter and spring, where future trends are consistent with past trends (i.e., winters will become wetter while springs drier).

For summer, future and historical trends differ slightly, where predicted SPI and SPEI depict clear negative trends. This, however, is in line with the moderate to severe dry summers of the last three consecutive years. In turn, SRI does not show a clear trend for the summer season. This could be due to uncertainties derived by the hydrological models on account of input data from the decadal members. In contrast to past trends, fall present positive trends depicted by all indices, which is inconsistent and should be further investigated. It should be noted that the low agreement in summer and fall do not compromise as much the usability of decadal predictions in the frame of reservoir management, as the critical seasons for filling up the reservoir are winter and spring.

There are slight differences between NASIM and SWAT, as these models were designed for different purposes. SWAT features a more complex description of runoff generation processes with respect to soil water and vegetation. NASIM focuses in turn on detailed urban drainage and complex hydraulic structures. Nevertheless, there is a general good agreement between both models for simulating discharge in the study area, showing that model uncertainty is relatively small. Uncertainties arise mainly from the input time series. This is an indication that selecting an appropriate hydrological model is not as critical as the quality of the input data.

Decadal predictions were validated for the overlapping period (2015-2020) by means of statistical tests. Test statistics confirmed an overall good agreement between ensemble members and observations. Statistically significant differences were mostly detected for summer discharge, where there is a significant overestimation of the models mean value with respect to observed mean. This suggests poorer skills of decadal members for summer discharge prediction. The effects of these uncertainties are transferred to both hydrological models, which show similar results. Therefore, future changes detected in summer should be interpreted with caution. The best agreement between observed and simulated discharge takes place in spring.

Mean seasonal variations forecasted for the period 2015-2024 do not present generally a strong change in hydrological variables, as shown by the graphical analysis. Decadal members predict higher evapotranspiration rates in summer as well as an increase in temperature of approx. 0.5° C by 2024, in agreement with past trends. The expected increase in temperature and evapotranspiration rates in winter is lower compared to summer, so a consequent impact on precipitation would not be as meaningful.

The spread of decadal prediction ensemble is as large as the inter-annual variability, making the detection of changes difficult. This indicates that the entire spread of model ensemble should be analyzed to not miss the potential variability in the predictions.

There is an underestimation of winter precipitation of the decadal members mean value compared with the validation period
(2015-2020). In turn, future SPI, SPEI, and SRI calculated with the corresponding ensemble means indicate that winters will become more humid with respect to average conditions. This demonstrates that the application of several methods to determine future trends provide more robust and reliable results.

Regarding simulated reservoir storage with different climate scenarios, decadal predictions present a better agreement in comparison to observed data than climate projections. Despite data uncertainties, all data sets show a negative trend by the end
of 2024: in none of the scenarios, the reservoir reaches the maximum storage level. Water stress at GDR by the end of 2024 is in conclusion not unlikely, and sustainable adaptation measures should be further considered.

**Data availability**

The hydro-meteorological time series used in this research can be accessed by contacting Maria Paula Lorza-Villegas (pla@wupperverband.de). Data from decadal predictions can be accessed by contacting Edmund Meredith
(edmund.meredith@met.fu-berlin.de) and afterwards via DECO plug-in: https://freva.met.fu-berlin.de/external/register/.

**Author contribution**

MPLV and RB set-up the models and performed the simulations and the analysis. MS provided the regional context from a hydrological point of view, considering the perspective of relevant stakeholders. TADB coordinated the research project. JR provided support with large data management and contributed to the analysis of results. MPLV prepared the paper with the
contributions of all co-authors.

**Competing interests**

The authors declare that they have no conflict of interest.



**Special issue statement**

This article is part of the special issue "Integrated assessment of climate change impacts at selected European research sites –

from climate and hydrological hazards to risk analysis and measures". It is not associated with a conference.

**Acknowledgements**

The authors would like to thank the Horizon 2020 project BINGO (http://www.projectbingo.eu/, last access: 30 December 2020), which offered us the possibility to gain insight of applications of climate data and their interpretation in the frame of reservoir management. Thanks to this research, relevant stakeholders were sensitized, and the Wupper Association is now

further prepared to take adaptation measures to face future water stress issues. Also, the authors would like to thank Thorsten Luckner, Alexander Löcke, Peter Nieland, and Daniel Heinenberg (Wupper Association) for their invaluable help and support in the development of this research.

**Financial support**

This research has been supported by the BINGO European H2020 project (grant no. 641739) and the Wupper Association

(Wupperverband).

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
