# Peer review of "Assessing short-term climate change impacts on water supply at the Wupper catchment area, Germany"

_Natural Hazards and Earth System Sciences, 2020_

## Referee Comment (RC1)

Review of NHESS-2020-429
Assessing short-term climate change impacts on water supply at the Wupper catchment
area, Germany.

The paper studies impact of climatic change on the runoff and potential to supply drinking water using short-term climate scenarios. As such it is interesting and relevant for the journal, and it is interesting to see the application of a different approach to providing climate scenarios. I think the results are nicely presented with good figures, and most of my comments and questions to the manuscript is related to the explanations on how and why things are done as they are.

I know this is an outcome from the BINGO project, and in that respect, I do have a couple of issues that came up while reading the paper. The major issue is related to the very frequent citations of BINGO reports. In some cases, this is of course perfectly ok but in other cases some of these results should be in the paper to improve the readability and prevent the reader for having to download a number of reports to find important information.
A minor issue is the textual references, e.g "I the frame of BINGO …" or similar language. From the intro we know that this is done in the BINGO project so that the text flow and language simplicity can be improved.

I have some questions related to the hydrological modelling:
- Why using both the NASIM and SWAT model in the project? There is a partly explanation of the difference of the model, but what is the benefit of using both in the Wupper catchment? Are they focused on different topics, and if so how do we interpret the results? From the text around line 145 it is not easy to understand.
- What strategy was used for calibration of the models?
- Did you consider uncertainty in the calibration?
- The calibration and validation if totally referred to a project report stored online. You could at least provide a summary of the calibration/validation, goodness of fit measures, difference in calibration results and what this means for the comparison of the simulations of the models. It would be particularly interesting to know how the models handled low flow periods.
- Can a hydrological model be physically and lumped? What do you mean with physically? I do not know NASIM, but isn´t SWAT semi distributed and in many process descriptions more conceptual than physical.

You do discuss and use the climate realizations both in the methods and in the results. The difference between these should be explained in the paper (or in supplementary material). This is too important for the understanding of the paper to be left in an online report. I think I understand that realisations 1, 7, 9 are max,min mean (line 159), but how large are the differences?

What is the basis for the claim that the decadal projections provide a more realistic assumption than the ensemble of RCP based scenarios?

I think a clarification of the text around line 75 is in order. So, the reservoir filling depends on spring precipitation and since spring precip now has shifted to summer this is no longer

possible. And in general (meaning in the past?) summer precipitation has not been important (since it previously was in the spring)?

At the start of the Data and methods section there is some basic drought information which should be moved to the introduction.

The text book information in the section on statistics could be more stringent. What method was used, when and the p-level used. All methods are standard so the meaning of $H_0$ and similar things are not needed.

The start of 4.1 is method material, could be moved. The use of Buchenhofen is also described before in the text.

I think I would have done the test of the overlapping climate predictions before I used them for analysis, but that is not very important.

Table 4: Is the runoff from NASIM or SWAT

Line 350-355: Why do you think the model perform like described here? Does this in any way related to the calibration/validation of the model, e.g. do you see the same pattern when you identify the model parameters? This is a reason to do a more detailed job in describing the model calibration, even if this exist in some online report.

I do also miss a short description of the reservoir model TALSIM-NG and particularly operational rules. A proper reference would also be in place. Any calibration or test of this model? What operational rules was used here, do they change in the future climate due to the changes in runoff observed?

Why was the RCP based models used here? What climate models was used to generate the RCP simulations, is it an ensemble of several?

In my opinion the conclusion is a bit long and repeats some findings presented before, to some extent it now works like a discussion.

What causes the large spread in simulations for the reservoir in figure 9? How does the volume of runoff in the scenarios compare to the observed?

Minor issues:
- Can you provide a proper reference for NASIM?
- On line 57 on the second page, could you say something more on the magnitude of changes in dry periods? Even if it is described before you could say how the rainy season has changed here.
- Line 69: What is meant with the "weather normal distribution" is it the weather normal period and if so 1961 – 1990?
- Line 79: "remained rather natural" could probably be simplified to "remained natural"

- Line 227: "was able to prove" is a bit awkward. The text "The non-parametric" could be removed. Should "less and equal" be "less"? You could just write "Both period show a significant positive trend (Mann-Kendall test, $p<0.05$)".
- Line 232: Isn´t it quite common to report very small p-values as $p<0.001$? Above you use $p<0.05$ and here the detailed decimal number, be consistent.
- Line 300 just repeats that you have used NSIM an SWAT.
- Line 330,
- Some places "less and equal" are used for p-values, shouldn´t that be "less" given your explanation in the method section?

---

## Referee Comment (RC2)

**Formal review to Lorza-Villegas et al. (2021), 'Assessing short-term climate change impacts on water supply at the Wupper catchment area, Germany'**

The manuscript submitted by Lorza-Villegas et al. (2021) deals with computations of different climate indices using decadal projections and hydrologic model simulations. The authors basically

(1) Give an overview over dry spell periods observerd at the Große Dhünn Reservoirs since the 1990s and illustrate response measures taken (Table 1)
(2) Analyse historical climate (rather temperature) data between 1991-2020 from the Buchenhofen station in Wuppertal (fig. 2)
(3) Calculate climate indices (SPI, SPEI, SRI) using measurement data from Neumühle on a seasonan scale and analyse the change between 1990 and 2020 (fig. 4)
(4) Derive Precipitation, Evapotranspiration and Temperature from decadal predictions between 2015 and 2024 and compare the frequency distribution to the ones derived from measurement data in 1990-2020 (fig. 5)
(5) Use two hydrological models to compute discharge from the decadal predictions of Precipitation, Temperature and Evapotranspiration and compare the observed distribution (1990-2014) to the prediction period (2015-2024) (fig. 6)
(6) For the overlapping period 2015 – 2020, the authors compare observed data with decadal predictions (P, ET and T) and calculated discharge (table 4, fig. 7)
(7) Calculate SPI, SPEI and SRI for the decadal prediction from 2015-2024 and analyse in the trends within the overlapping validation period (2015-2020) compared to the trend of the historical period (1990-2020) (3)
(8) Simulate Storage Volume of GDR using another hydrological model and different climatic forcing (relaization 1 and 9, rcp45 and rcp 85) between 2015-2024 and compare those simulations with observations (2015-2020)

The interesting part about the study is, in my opinion, the aim to relate decadal climate forecast results (generated in the framework of an applied EU Project (BINGO)) to local obeservations, use it for further discharge modelling and trying to derive information usable for water management. The innovative aspect could, thus, be a judgment of the usefulness of such a data set for stakeholders in the region, an aspect that in my opinion is relevant and worth being published in an international journal dealing with these sorts of impact assessements.

However, the current manuscript reads more like a project report than a scientific paper. In my opinion, the following improvements should be made:
- The research focus should be clearified
- Many topics are opened by the authors but only scratched at the surface such as for instance the topic of decadal and ensemble prediction techniques, advantages of climate indices or model uncertainties (which is quite often mentioned but not really defined or quantified).
- The language could be more precise and should be improved.
- There is a lack of information about the performance of the hydrological models compared to observations, and not enough information on the skill of the decadal predictions is presented.
- Significance levels are missing on some of the trend analysis and, in my opinion, the authors sometimes tend to over-interpret some of the results

- The conclusions are rather a discription of model outcomes. However, a discussion about value and usability of such data for a local water association is missing – which has been, according to the authors, the main goal of the BINGO project.

In more detail, I have the following remarks

**1) Introduction:**

Even though the general outline of the study is given, a clear problem statement, research questions or hypotheses are missing. The introduction is somewhat underreferenced, and not much detail is given on state-of-the art literature about the topic. This would help to identify a research gap which the authors aim to close with their study. It is not discussed later in the study if the mentioned „main goal of BINGO" (to give practical tools to local water authorities) has been reached.

**2) Study area:**

For the reader it is a bit irritating that parts of the climate description of the area is given in this section (including table 1 showing an overview on dry spell and response measures), whereas more detailed data is given later in fig. 2. Furthermore, table 1 is somewhat isolated within this study. Relating „dry spells" to periods of GDR < 35 mrd m³ is probably only valid if outflow regulation of the reservoir is constant and if no changes occured in the catchment. Furthermore, it remains unclear why the very dry years (2018-2020) obviously did not lead to a dry spell event at GDR.

The information about projected changes in precipitation (line 85) is related to Germany as a whole and therefore probably not that helpfull - especially since more detailed information is given later.

**3 Data and methods:**
**3.1 Data**
**3.1.1 Observed data**

It should be claryfied what the different data sources are used for (i.e. the temperature data from Wuppertal). Where is sunshine and wind speed coming from? Maybe a table that relates the observed data to the result and discussion sections would help. The sentence *GDR storage time series are … used for comparison with simulated volume with different scenarios* is an example of being not very specific (what scenarios?)

**3.1.2 Decadal predictions**

I expected some information on the derivation of decadal projections, as it appears to be a central aspect of the paper (for instance on which models the predictions are based on, about the spatial and temporal resolution of the predictions, how they are derived or how the ensembles are set up, if there is any postprocessing such as downscaling or bias correction applied and so on).

**3.1.3 Runoff and reservoir simulations**

Nasim and Talsim are probably not well known models in the international community. Thus, both could be presented in some detail. The reason why a third model (Talsim) is used for the volume calculation remains unclear.

It is claimed that the models have been calibrated and different metrics have been used. I recommend to present some of the metrics and give some more information on model set up (i.e model time step, spatial resolution calibration period etc.).

Line 144: NASIM and SWAT are *no* physical hydrological models.

Line 154: It is not possible to compare simulations to observations up to 2024.

Line 157: Comparison to RCP4 an RCP8.5 and the use of individual decadal members should be adressed in the introduction. In this section, it remains unclear why this is done.

**3.2 Statistical tests**
Line 165: *In this manner, it is possible to evaluate the skill of forecast series as a way of quantifying uncertainty*: It should be mentioned how uncertainty is quantified, otherwise ‚uncertainty' rather seems to be used like a buzzword.

**3.3 Drought and climate indices**
It would be interesting to know why indices are used instead of model outputs directly (such as precipitation, temperature and runoff). The „number of summer days" as an index is only used for observations (not for predictions) and therefore does not provide any further information.

**4 Results and discussion**
**4.1 Historical climate analysis**
Line 221: *For this analysis, Buchenhofen station (located in the city of Wuppertal) was selected as it provides continuous temperature time series since 1948.*
What exactly is the benefit of deriving additional information from the analysis of Buchhofen Temperature and summer days in this study ? It is well known that temperatures did rise during the last 50 years. However, if a climate analysis is performed, why is there no discussion about i.e. precipitation?

Line 236: *Particularly, the highest measured temperature was recorded in the summer of 2018, during which an extreme flood event caused by heavy rain took place in the city of Wuppertal.*
Should that suggest a causality between highest temperatures and most extreme flood events?

Line 239: *The temperature rise has led to an increase in convective heavy precipitation events, particularly in summer, further impacting the monthly and seasonal precipitation regime.*
You should give some data here supporting this statement. It is not necessarily true that an increase in heavy precipitation events (on short time scales) significantly impacts monthly or seasonal precipitation amount (if on the other hand the number of dry days increase as well).

Line 240: *This is consistent with Hartmann et al., 2013, ...*
Relating a general (may be global) IPCC statement to the local Wupper area situation appears questionable.

Line 241: *In conjunction with the subsequent greater water demand and rise in AED, river streamflow (i.e., inflow runoff to GDR) will continue to be affected in the future.*
This statement is again rather unspecific (will be affected in what way )

Figure 3: The benefit of the given information is not clear to me.

Figure 4. The limitation of the discussion of these figures is - although there was a section on statistical tests in 3.2 - that no information is given on the significance of the trends. The advantage of looking on indices should be discussed.

Line 259:*Spring: .. SPEI in turn shows that springs have become drier due to higher evapotranspiration rates caused by increasing temperatures*
Since no increase in spring temperature has been shown, it might also be due to higher incoming radiation or wind speed

*…indicating that in spring, SRI is more responsive to precipitation shortages*
More responsive compared to what? why?

Line 261: *This suggests that the decrease of precipitation in spring overtime has a greater impact on runoff generation on account of other components of the hydrological cycle (e.g., infiltration and groundwater recharge).*
I do not really understand this sentence (greater impact compared to what? why?)

Line 267: *Even though there is no clear SPI or SPEI trend, a sequence of moderate to severe dry summers have occurred in the last three years.*
What do you want to say with that ? Other 3-year periods have not been discussed ….

Line 269: *It can be then inferred that the decrease in SRI trend in spring is mainly on account of evapotranspiration rates, whereas in summer, the negative SRI trend takes place due to catchment wetness by the end of the spring season.*
Why shoud the negative SRI trend be (only) due to the situation at the end of spring? As you mentioned there could have been a shift toward more intense precipitation in summer (and more drought days) - which together could lead to less runoff. Probably, a 3 month sum of rainfall is not a very good proxy for the wetness state.

Line 271 *Fall: all three indices show a negative trend, whereas SRI presents the strongest slope. Following the same pattern as in summer, this can be the result from initial drier conditions at the beginning of fall due to decreasing SRI values at the end of the precedent season.*
From Figure 4, the difference in slope between SPI and SRI seems to be insignificant.

**4.2 Future Changes in precipitation, temperature, ET0 and discharge**

Line 281: *The historical reference period is defined here as the time interval from 01 January 1990 to 31 December 2014.*
In previous sections there was a control period (1961 – 1990) and an ongoing climate normal (1991-2020). The reference period beeing now only up to 2014 means ignoring the dry years 2018-2020 which is sort of inconsistent.

Figure 5: Information about the different boxplot (a – c) should be given in the figure caption

Line 289: *Strongest reductions are estimated in the winter month ...*
This should be discussed – as far as I know most scenarios predict increasing precipitation amounts for winter in western Germany. Is that not in contradiction to the positive SPI trend shown for winter in 1990-2020 (figure 4) ?

Line 298: *Winter months show a slight reduction in ETo, however, ETo does not play a significant role in winter in any case*
That is true, but what is the reason for the evaporation decrease in winter, spring and fall – even though temperatures are rising? This somehow contradicts the discussion of the historical trends.

What I miss here is a comparison of the seasonal prediction for the historical period (reanalysis). Showing that the decadal predictions are capable of reproducing past climate observations would give much more confidence to any future impact analysis using hydrological models.

Line 300 ff  Hydrological modelling (and figure 6)
Observed discharge from 1990 to 2014 is compared to simulated discharge (Q) by NASIM and SWAT for the period 2015 to 2024. For any interpretation it would be relevant to see how the models compare to observation in 1990 – 2014

Line 304: *Both models show similar trends in their runoff estimations and predict decreasing discharge rates in winter (approx. 0.1 to 0.2 m³ s-1) and increasing discharge in summer and fall (approx. 0.1 m³ s-1 and 0.05 m³ s-1, respectively)*
No information is given about the significance of these „trends" (note that you are talking about highly aggregated data)

Line 307: *Even though the predictions of both models for spring lie within the interquartile range of the historical data, a significant increase or decrease compared to past conditions cannot be stated.*
I don't understand the relation between the interquartile range of the historical data compared to any future trend

Line 310: *there is a general good agreement between both models, showing that model uncertainty is relatively small, and the use of hydrological model ensemble is valid for further analysis*
I am not sure it this conclusion is valid from the information presented here. No uncertainty quantification of the models has been done. Showing a similar distribution for a three month mean on a decadal time scale does not mean that both models do not end up with very different results for individual years or events (and thus large uncertainties).

Line 311: *This is an indication that selecting an appropriate hydrological model is not as critical as the quality of the input data.*
Do you mean the input data quality is not sufficient for differences in hydrological models to become relevant ?

Line 314: *For this reason, ensemble studies are often done with the aim to provide an ensemble mean value of climate variables*
In my opinion the ensemble benefit is in giving uncertainty ranges. Your figure 6b illustrates that focussing on the ensemble mean causes a severe loss of information.

Line 324: *As with decadal predictions, where ensemble mean aims to reduce model uncertainties ... it seems reasonable to apply the same principle to resulting discharge by two different hydrological models*
and
Line 326: *Past studies have shown that uncertainty can be reduced even with simple ensemble techniques such as arithmetic mean ... Moreover, hydrological model ensemble can yield better results than each individual model or the best one of them*
Averaging neither does necessarily give you a any better results, nor does it reduce any uncertainty. However, you could show that by comparing hydrological model results to past observations.
The idea behind an ensemble is to take into account the conceptual uncertainties of your applied models, i.e. by varying state parameters within the range of their uncertainty. In this sense, just taking two models is not really an ensemble.

Line 330: *Today, after the creation of the decadal predictions ... forecast with observed data for almost six overlapping years*
A comparison to observed data would have been expected earlier in this study (note that this is the section 4.3 Future Changes in precipitation, temperature, ET0 and discharge) and not only for 6 years

Line 336: *The hydrological model ensemble of NASIM and SWAT was used for discharge ...*
I assume that the models are driven by the (historic) decadal predictions. Again, it would be interesting to see how they perform using observed data

Line 338: *Test statistics confirm an overall good agreement between ensemble members and observations*
What ensemble „members" do you mean (also unclear in table 4): Earlier in this study the focus was on ensemble mean.
The numbers for a 3 month seasonal mean with a large overestimation in summer and fall and a clear underestimation in winter indicate, that the simulations do not reproduce the mean discharge hydrograph.

*Table* 4 Why is there a different test for Precipitation in Winter compared to the other seasons?

Line 355: *The best agreement between observed and simulated discharge takes place in spring. This is an important finding for the water management and planning in this region.*
What exacly is the use of that finding for water mangement - only trust decadal predictions in spring?

**4.3 Future changes in drought indices and reservoir storage**

Figure 8: These are just 9 values. Adding trend lines and much interpretation is critical

Line 381 ff:  What are realizations 1 and 9? what is realization 7? Why are now rcp scenarios compared?

Line 385: *Given the good agreement between observed and simulated reservoir storage for the validation period of 2015 to 2020, water stress by the end of 2024 is not an unlikely scenario*
How does that match with figure 8 (indicating an above average SRI in 5 of the last 6 season before end of 2024)?

**5) Conclusions**
Generally this is more a repitition of the discusion section than a clear conclusion. Based on the introduction, I would expect a (critical) judgement if decadal predictions provide information reliable enough for a local water authority, as well as the benefit of using climate indices compared to directly look on the variables (P, ET, Q)

Line 420: *It should be noted that the low agreement in summer and fall do not compromise as much the usability of decadal predictions in the frame of reservoir management, as the critical seasons for filling up the reservoir are winter and spring.*
In my opinion this is not critical enough. Could you trust the predictive ability of a model that is not capable of reproducing the mean hydrograph on a seasonal timescale ?

Line 423: *There are slight differences ...  h*howing the model uncertainty is relatively small:
In fact no uncertainty analysis or quantification has been done.

In summary, I recommend the manuscript to be published if major revisions are performed.